# Familial Aspects of Mammographic Density Measures Associated with Breast Cancer Risk

**DOI:** 10.3390/cancers14061483

**Published:** 2022-03-14

**Authors:** Tuong L. Nguyen, Shuai Li, James G. Dowty, Gillian S. Dite, Zhoufeng Ye, Tu Nguyen-Dumont, Ho N. Trinh, Christopher F. Evans, Maxine Tan, Joohon Sung, Mark A. Jenkins, Graham G. Giles, Melissa C. Southey, John L. Hopper

**Affiliations:** 1Centre for Epidemiology and Biostatistics, The University of Melbourne, Melbourne, VIC 3010, Australia; nguk@unimelb.edu.au (T.L.N.); shuai.li@unimelb.edu.au (S.L.); jdowty@unimelb.edu.au (J.G.D.); gillian.dite@gtglabs.com (G.S.D.); zhoufengy@student.unimelb.edu.au (Z.Y.); nhut.trinh@unimelb.edu.au (H.N.T.); cfevans@unimelb.edu.au (C.F.E.); m.jenkins@unimelb.edu.au (M.A.J.); graham.giles@cancervic.org.au (G.G.G.); 2Centre for Cancer Genetic Epidemiology, Department of Public Health and Primary Care, University of Cambridge, Cambridge CB1 8RN, UK; 3Precision Medicine Group, School of Clinical Sciences at Monash Health, Monash University, Melbourne, VIC 3168, Australia; tu.nguyen-dumont@monash.edu (T.N.-D.); melissa.southey@monash.edu (M.C.S.); 4Genetic Technologies Limited, Melbourne, VIC 3065, Australia; 5Department of Clinical Pathology, The University of Melbourne, Melbourne, VIC 3010, Australia; 6Electrical and Computer Systems Engineering Discipline, School of Engineering, Monash University Malaysia, Bandar Sunway 47500, Malaysia; maxine.tan@monash.edu; 7School of Electrical and Computer Engineering, The University of Oklahoma, Norman, OK 73019, USA; 8Division of Genome and Health Big Data, Department of Public Health Sciences, Graduate School of Public Health, Seoul National University, Seoul 08826, Korea; jsung@snu.ac.kr; 9Cancer Epidemiology Division, Cancer Council Victoria, Melbourne, VIC 3004, Australia

**Keywords:** breast cancer, cirrocumulus, cumulus, familial risk ratio, heritability, mammogram risk score, mammographic density, OPERA

## Abstract

**Simple Summary:**

*Cumulus*, *Cumulus-percent*, *Altocumulus*, *Cirrocumulus*, and *Cumulus-white* are mammogram risk scores (MRSs) that predict a woman’s risk of breast cancer based on mammographically dense areas when defined by different levels of brightness. We measured these MRS for 593 monozygotic (MZ) and 326 dizygotic (DZ) female twin pairs and 1592 of their sisters. We estimated how much these MRSs were correlated in relatives (*ρ*), how much of the differences between women were due to genetic factors (heritability), and how much these MRS explained why breast cancer runs in families. The *ρ* estimates ranged from: 0.41 to 0.60 for MZ pairs, 0.16 to 0.26 for DZ pairs, and 0.19 to 0.29 sister pairs, respectively. Heritability estimates were 36% to 69%. Genetic factors explain most of why twins and sisters are similar in their MRS, and these genetic factors explain one-quarter to one-half as much breast cancer risk as to the current best genetic risk score.

**Abstract:**

*Cumulus*, *Cumulus-percent*, *Altocumulus*, *Cirrocumulus,* and *Cumulus-white* are mammogram risk scores (MRSs) for breast cancer based on mammographic density defined in effect by different levels of pixel brightness and adjusted for age and body mass index. We measured these MRS from digitized film mammograms for 593 monozygotic (MZ) and 326 dizygotic (DZ) female twin pairs and 1592 of their sisters. We estimated the correlations in relatives (*r*) and the proportion of variance due to genetic factors (heritability) using the software FISHER and predicted the familial risk ratio (FRR) associated with each MRS. The *ρ* estimates ranged from: 0.41 to 0.60 (standard error [SE] 0.02) for MZ pairs, 0.16 to 0.26 (SE 0.05) for DZ pairs, and 0.19 to 0.29 (SE 0.02) for sister pairs (including pairs of a twin and her non-twin sister), respectively. Heritability estimates were 39% to 69% under the classic twin model and 36% to 56% when allowing for shared non-genetic factors specific to MZ pairs. The FRRs were 1.08 to 1.17. These MRSs are substantially familial, due mostly to genetic factors that explain one-quarter to one-half as much of the familial aggregation of breast cancer that is explained by the current best polygenic risk score.

## 1. Introduction

Conventionally, mammographic density refers to the white or bright areas on a mammographic image. These regions are critical for breast screening because they are used by radiologists to identify potential cancers, but at the same time, they can make it difficult to detect existing cancers (i.e., they have a masking effect). The extent of mammographically dense regions on a breast image is also associated with a woman’s risks of being diagnosed with breast cancer: (i) at the time of screening, (ii) in the interval before the next regular screen, and (iii) at a future screen [1].

There are many paradoxes involved here, not the least being that because masking increases the risk of missing cancer at mammographic screening, it reduces, not increases, the incidence of screen-detected breast cancer [1]. Another complication is that the mammographic dense area, especially when considered a percentage of the total breast area, decreases with age and with increasing weight and body mass index (BMI) [2]. However, breast cancer risk increases with age and—at least post-menopause—with weight and BMI. The latter issues can be addressed by adjusting analyses for age and BMI, but this is often reduced to a footnote in Tables and not appreciated when these adjustments are in effect ignored (due to imprecise language) when the results are interpreted.

Therefore, we define a mammogram risk score (MRS) as the residual of a mammographic measure (transformed to approximate normality if needed) after making an adjustment for age and BMI based on the analysis of controls or another sample to represent the population and then standardizing it to have mean = 0 and variance = 1. This approach allows the simple estimation of the odds ratio per adjusted standard deviation (OPERA), where adjusted standard deviation refers to the standard deviation of the adjusted measure, not of the unadjusted measure [3]. The log(OPERA) is equal to Δ = the difference between cases and controls in the mean of the MRS and is related to the area under the receiver operating characteristic curve (AUC) by the formula AUC = φ(Δ/2^½^) [4].

Conventionally, mammographic density is defined at the area of the breast covered by white or bright regions and can be measured using the computer software CUMULUS [5]. We name the raw density measure based on this definition Cumulus density, and for the MRS obtained by transforming, adjusting, and standardizing Cumulus density, we name it *Cumulus*.

Over the last few years, we introduced other mammographic density measures based on defining density at, in effect, higher pixel brightness thresholds [6,7,8,9]. We name Altocumulus density the area of density defined as the bright regions, and Cirrocumulus density, the area of density defined as the brightest regions. *Altocumulus* and *Cirrocumulus* are the transformed, adjusted, and standardized MRSs based on these two measures, respectively. Therefore, *Cumulus* is the MRS for Cumulus, *Altocumulus* is the MRS for Altocumulus. and *Cirrocumulus* is the MRS for Cirrocumulus.

From a series of studies of Australian and Korean women, we consistently found that *Altocumulus* and *Cirrocumulus* provide more information on risk than *Cumulus* [6,7,8,9,10]. Furthermore, the risk association for *Cumulus* attenuates and often becomes null after adjusting for *Altocumulus* or *Cirrocumulus* [1]. An exception is the risk of interval breast cancer, for which the conventional Cumulus measure (as a percentage) is a consistent predictor of risk, even without adjusting for age and BMI (likely due to its role in masking) [10]. Nevertheless, all three measures have a role to play in predicting breast cancer risk.

The potential role of familial, if not genetic, factors in explaining variation in the MRS has been considered for more than two decades. In 2001, a joint Australian–Canadian study of *Cumulus-percent*, based on Cumulus as a percentage of the total breast area and adjusted for age and BMI, found that the correlations for monozygotic (MZ) twin pairs were about 0.6, whereas they were about one-half of this (about 0.3) for dizygotic (DZ) twin pairs. The twin pair correlations and total variances did not differ between countries. Given the strong correlation of >0.9 between *Cumulus* and *Cumulus-percent*, this is consistent with additive genetic factors explaining about 60% of the variation in both *Cumulus* and *Cumulus-percent* [11]. Based on Hopper and Carlin [12], we predicted that these conventional mammographic density MRSs explain, in a statistical sense, around 10% of the overall familial risk of breast cancer.

We more critically addressed the causes of familial variation in *Cumulus* by enlarging the Australian twin data set and recruiting sisters of these twin pairs [2]. We found similar results with the larger twin sample. We also found no difference in the correlation between DZ pairs and sister pairs (both share, on average, half their genetic variants). Under the equal environments assumption of the classic twin study, this is consistent with genetic factors fully explaining the familial correlations, and there is no evidence that non-genetic factors shared by twins and sisters have a detectable influence on variation.

In this study, we aimed to estimate the familial correlations of all the MRS and again try to decompose the variance into genetic and non-genetic components. We used the enlarged twins and sisters data set to consider MRS based on *Cumulus*, *Cumulus-percent*, *Altocumulus,* and *Cirrocumulus*, but this time the MRS were measured by a different set of measures.

Here we introduced a new mammogram risk score called *Cumulus-white*, based on transforming, adjusting, and standardizing the difference between the raw Cumulus density measure and the raw Altocumulus density measure. This new measure represents the white but not bright areas on a mammogram.

We again considered the extent to which the familial aspects of these MRSs explain the familial aggregation of breast cancer based on their published risk gradients. We also compared the proportion of familial aggregation in breast cancer explained by these MRS with that of the latest polygenic risk score (PRS) based on 313 single-nucleotide polymorphisms (SNPs) [13].

## 2. Materials and Methods

### 2.1. Sample

We used data from the Australian Mammographic Density Twins and Sisters Study [2]. Briefly, female twin pairs aged 40–70 years without a prior diagnosis of invasive breast cancer were recruited through Twins Research Australia. Participating twins completed a questionnaire and gave permission to access their mammograms. They were also asked to seek permission from any eligible sisters to be invited to participate in the study. We recruited 3430 twins and sisters from 1578 families, including 593 MZ and 326 DZ twin pairs and 1592 non-twin sisters. All participants gave written informed consent, and the study was approved by the Human Research Ethics Committee of the University of Melbourne.

### 2.2. Risk Factor Questionnaire

Telephone-administered questionnaires were used to record demographic information and self-reported weight, height, smoking history, alcohol consumption, reproductive history, onset and cessation of menstruation, use of oral contraceptives, use of hormone replacement therapy, and personal and family history of cancer. For twin pairs, zygosity was determined from genome-wide analysis study data [14].

### 2.3. Mammographic Density Measurements

All available episodes of mammograms were retrieved with the participants’ written consent, mostly from Australian BreastScreen services, but also from private clinics and private hospitals. We also retrieved mammograms from the participants themselves. The craniocaudal views for left and right breasts were selected and digitized using the Lumysis 85 scanner at the Australian Mammographic Density Research Facility. For each woman, the most recent right breast craniocaudal view was selected for mammographic density measurement, and the left breast craniocaudal view was selected if the right breast mammogram was missing or unavailable. Mammographic measurements of total area and dense area were performed using CUMULUS 4.0, a computer-assisted thresholding technique, after randomization and blind to information, by three independent measurers (TLN, HNT, CFE) with high repeatability; see for example [15]. Dense area (DA) and percent dense area (PDA), defined as DA divided by total area expressed as a percentage, were measured. We also subtracted the Altocumulus DA measure from the Cumulus DA measure to create Cumulus-white, which refers to the white, not bright, areas on a mammogram.

### 2.4. Statistical Methods

The Box–Cox procedure was used to test the normality of the distributions of the mammographic density measures and, if necessary, select an appropriate power transformation. We present the MRSs based on both the absolute density measure and the percent density measure, *Cumulus* and *Cumulus-percent*, even though these are highly correlated, due to both measures often being reported in the literature. For Altocumulus density and Cirrocumulus density, we reported only the MRSs based on the absolute density measures. To create *Cumulus*, the absolute density measure was cube root transformed, while for *Cumulus-percent*, the percent density measure was square-root transformed. To create *Altocumulus*, the absolute density measure was cube root transformed and to create *Cirrocumulus*, the absolute density measure was log-transformed. For *Cumulus-white*, the difference between the Cumulus and Altocumulus absolute density measures was log-transformed. All transformed measures were adjusted for age and the inverse of BMI, and the residuals were standardized to generate unit variance to create the MRS. This was performed using linear regression applied to data for controls only to estimate the two regression coefficients and the constant term and thereby the expected value as a function of age and BMI and the variance of the residuals for controls. For both cases and controls, their observed transformed values were then subtracted from their expected values based on the regression analysis of controls above to derive the residuals, which were then divided by the variance of the residuals for controls. We previously found that breast cancer risk factors measured by questionnaire, other than age and BMI, explained, at most, a slight percentage of variance in the conventional density measure, *Cumulus*, and this was trivial compared with the variance explained by age and BMI [2]. We also found that the same applies to the new density measures, *Altocumulus* and *Cirrrocumulus*. These statistical transformations were conducted using R version 4.0.2 software [16].

We applied a multivariate normal model for pedigree analyses fitted using the software FISHER, with statistical inference based on asymptotic likelihood theory, to estimate the correlation between pairs of relatives and to fit the variance components models [17,18]. This approach assumes that, after adjusting the mean for specified measured variables, the residuals follow a multivariate normal distribution with a covariance structure that can be flexibly parameterized. The approach allows the estimation of correlations separately for MZ and DZ twin pairs and or for sister pairs (including a twin and her non-twin sister).

We also fitted models estimating independent genetic and environmental components of variance to represent additive genetic factors (A), environment factors shared by twins and sisters (C), and individual specific environmental factors and measurement error (E), where A + C + E = total residual variance (V). MZ pairs share all their genes while DZ pairs and sister pairs share, on average, half their genes, so the correlation in additive genetic factors is 1.0 for MZ pairs and 0.5 for DZ and sister pairs [19]. Under the assumption of the classic twin study, that the effects of non-genetic (i.e., environmental) factors shared by twins and sisters are independent of zygosity and the same for twins and sisters, the correlation between a pair will be (2φ_ij_A + δC)/V where 2φ_ij_ = 1 if MZ else 0.5 and δ = 1 for all pairs.

We also fitted a model in which C was defined as an MZ pair-specific shared non-genetic environmental factor, which we refer to as C_MZ_, so that the correlation between a pair will be (2φ_ij_A + δC_MZ_)/V, where δ = 1 for MZ pairs, otherwise it is 0. Finally, we also fitted a purely genetic model which did not include any shared environmental factors but instead included a term D to represent dominance genetic factors such that the correlation between a pair is (2φ_ij_A + δD)/V, where δ = 1 if MZ, otherwise it is 0.25 [19]. Note that, for all models, the sum of the variance components is 1; therefore, it is not necessary to report the E estimates. Note that heritability, the proportion of variation explained by genetic factors, is given by A, or A + D, and the proportion of variation due to familial factors is A + C or A + D, respectively.

We estimated the proportion of familial aggregation in breast cancer explained by the familial correlations in the MRS and PRS by reference to Table 1 of Hopper and Carlin [12], in which RR is the inter-quartile risk ratio and *ρ* is the pair correlation. These results are consistent with those of Aalen [20] and the Supplemental Material of Clayton [21]. Based on the inter-quartile risk ratio of the risk score and its correlation between MZ twin pairs, we estimated the familial risk ratio (FRR) for MZ pairs.

## 3. Results

### 3.1. Correlations between Relatives

Table 1 show that the summary statistics for MZ twins, for DZ twins, and for non-twins did not differ substantially from one another.

Table 2 show that, numerically, all the MRS correlations were higher for MZ pairs than for the other two categories of relatives and that the correlations did not differ greatly between DZ pairs and sister pairs; see Model 1. Comparison of the (independent) MZ and DZ pair correlations and their standard errors shows that they differ from one another for every MRS (all *p* < 10^−4^). Model 2 constrains the DZ pair and sister correlations to be the same, and comparisons of the log-likelihoods with those of Model 1 show that for no MRS, there is evidence that the DZ pair and sister-pair correlations differed.

Model 3 constrains all pair correlations to be the same, and comparisons of the log-likelihoods with those of Model 1 show that there is highly significant evidence that this model is not consistent with the data; a similar statement applies to comparisons with Model 2. Therefore, we conclude that the MZ pair correlations are greater than those for the DZ pairs and sister pairs, which are similar to one another. Model 4 constrains the MZ pair correlation to be twice the correlation for DZ and sister pairs combined, and the comparison of the log-likelihoods with those of Model 2 shows that, except for *Cirrocumulus*, there was at least nominally significant evidence that the MZ correlations were more than twice the corresponding correlations for DZ and sister pairs combined (all *p* ≤ 0.05). The best-fitting models were Model 2 for *Cumulus*, *Altocumulus*, and *Cumulus-white*, and Model 4 for *Cumulus-percent* and *Cirrocumulus*.

### 3.2. Variance Conponents

We considered the extent to which the familial correlations accorded with different variance components models; see Table 3. Model 1 shows that when the ACE model was fitted, the C estimate was zero (the lower bound for a variance component), and the additive genetic components were essentially the same as the corresponding MZ pair correlations in Table 1. Model 2 shows that when C was replaced by C_MZ_, the estimates of C_MZ_ were all positive, and nominally so, except for *Cirrocumulus*. These significant C_MZ_ estimates ranged from 0.10 to 0.15, and the corresponding A estimate was reduced by a similar amount when compared with their estimates using Model 1.

When the ADE model was fitted, the estimates of A and D (all standard errors ~0.10) were: 0.41 and 0.30 for *Cumulus*; 0.43 and 0.20 for *Cumulus-percent*; 0.34 and 0.27 for *Altocumulus*; 0.32 and 0.08 for *Cirrocumulus*; and 0.30 and 0.31 for *Cumulus-white*, respectively. The correlations between the A and D estimates were approximately −0.98. These model fits had the same log-likelihoods as their corresponding ACE fits.

### 3.3. Explaining Familial Aggregation

The inter-quartile risk ratio for the MRSs as predictors of breast cancer are in the range of 2.5 for *Cumulus*, consistent across studies, and up to 5.0 for *Altocumulus* and *Cirrocumulus* from some studies [1]. We found above that the MZ pair correlations are about 0.6 for the former and 0.4 for the latter. Based on Table 1 in Hopper and Carlin [12] (see Statistical Methods), we predict that the FRR for breast cancer generated by the MRSs is about 1.08 for *Cumulus*, and up to 1.12 or 1.17 for *Altocumulus* and *Cirrocumulus*, respectively.

In comparison, for the current best PRS, which has an inter-quartile risk ratio of 3.5 [13] and is perfectly correlated in MZ twin pairs, the FRR for breast cancer generated is about 1.3. Therefore, on the log(FRR) scale, the MRSs individually explain about one-quarter to one-half as much of the familial aggregation of breast cancer as explained by the PRS.

## 4. Discussion

The familial correlations in the MRSs are substantial. In particular, the correlations for MZ twin pairs are in the range of about 0.4 to 0.7, which means that about half or more of the variance of these MRSs is familial. The MZ pair correlations are clearly greater than those for DZ pairs and sister pairs, while the latter correlations are generally similar to one another. Therefore, the correlations are related to the genetic similarity of relative pairs.

In terms of heritability, under the equal environments assumption of the classic twin model, extended to become that all twins and sister pairs share, to the same extent, all of the non-genetic factors that determine variation in the MRS, genetic factors would explain about 39% to 69% of the total variance of the MRS; see Model 1 of Table 2. If we take into account that, except for *Cirrocumulus* and *Cumulus-white*, the MZ correlation is significantly more than twice the correlation for DZ pairs and sister pairs combined, the heritability estimates reduce to 36% and 56%. The strong familial nature of these MRS (most specifically, the MZ pair correlations of 0.41 to 0.71) means that they explain a proportion of familial aggregation of breast cancer, as explained in Hopper and Carlin [12]. The familial risk ratio (FRR) is the relative risk associated with having an affected relative of a given type. For breast cancer, the FRR depends strongly on the age at diagnosis of the affected relative and the age of the at-risk woman, as well as with the relationship(s) with and the number of affected relatives [22], and even cancers other than breast among relatives. In particular, for MZ twin pairs, the FRR is six if the affected twin was diagnosed before the age of 50 years, reducing to less than three if the affected twin was diagnosed after the age of 60 years [23]. MZ twin pairs correlations set a natural upper limit to the role of genetic factors in causing disease concordance in relatives. If the overall genetic risk is considered a global genetic risk score in the same sense as we are considering the mammographic density measures to be MRS, then the inter-quartile risk ratio across the genetic risk score must be about 20 or more depending on the age of the at-risk woman [12].

We predicted that the FRR generated by the MRSs are about 1.08 to 1.17 and therefore explain about one-quarter to one-half of as much of the familial aggregation explained by the current PRS. In the typical screening age range of 50 to 70 years, the PRS explains about one-quarter of all familial aggregation and the MRS up to another one-eighth.

The question then remains regarding the genetic, and perhaps non-genetic, familial factors that determine these mammographic density MRS and how they relate to the PRS and its components. We will address this issue in a concurrent paper in this series [14].

## 5. Conclusions

There are substantial familial correlations in the MRSs that could have a genetic cause and explain about one-quarter to one-half of the familial aggregation of breast cancer that is explained by the current PRS.

## Figures and Tables

**Table 1 cancers-14-01483-t001:** Characteristics and measures of study sample by twin status for monozygotic (MZ) and dizygotic (DZ) twins. BMI = body mass index.

Characteristics and Measures	Total (*n* = 3430)	MZ Twins (*n* = 1186)	DZ Twins (*n* = 652)	Non-Twins (*n* = 1592)
Breast cancer risk factors, mean (standard deviation)				
Age (years)	53.7 (8.4)	54.1 (8.2)	53.5 (9.0)	53.4 (8.4)
BMI (kg/m^2^)	26.2 (5.3)	25.7 (4.9)	26.5 (5.2)	26.5 (5.5)
Mammogram measures, median (inter-quartile range)				
Cumulus	27.4 (17.0–41.2)	28.3 (17.7–41.1)	28.8 (18.2–42.7)	26.4 (15.6–40.5)
Cumulus-percent	28.1 (15.1–41.8)	29.9 (16.8–43.1)	28.4 (15.5–42.1)	26.9 (13.5–40.7)
Altocumulus	10.9 (6.4–16.4)	11.1 (6.7–16.1)	11.0 (6.2–16.9)	10.7 (6.3–16.4)
Cirrocumulus	1.7 (0.8–3.2)	1.6 (0.7–3.0)	1.6 (0.7–3.2)	1.8 (0.8–3.5)
Cumulus-white	15.9 (9.4–25.3)	16.6 (10.1–19.8)	16.9 (10.7–20.3)	15.1 (8.5–24.3)

**Table 2 cancers-14-01483-t002:** Correlations in mammogram risk scores (standard errors in parentheses) for categories of relatives under unconstrained (model 1) and constrained models (model 2: DZ = sister; model 3: MZ = DZ = sister; model 4: DZ = sister = 0.5MZ) with *p*-value for designated comparison of model fits based on log-likelihoods.

Relative Pairs	*Cumulus*	*Cumulus-percent*	*Altocumulus*	*Cirrocumulus*	*Cumulus-white*
Model 1					
MZ twin pairs	0.70 (0.02)	0.63 (0.02)	0.61 (0.02)	0.41 (0.03)	0.61 (0.02)
DZ twin pairs	0.25 (0.06)	0.26 (0.05)	0.20 (0.05)	0.16 (0.05)	0.22 (0.05)
Sister pairs	0.29 (0.03)	0.27 (0.03)	0.25 (0.03)	0.19 (0.03)	0.23 (0.03)
Log-likelihood	1464.084	−1522.568	−1535.675	−1637.656	−1522.304
Model 2					
MZ twin pairs	**0.70** **(0.02)**	0.63 (0.02)	**0.61** **(0.02)**	0.40 (0.03)	**0.61** **(0.02)**
DZ twin pairs	**0.28** **(0.02)**	0.27 (0.02)	**0.24** **(0.02)**	0.18 (0.03)	**0.23** **(0.02)**
Sister pairs	**0.28** **(0.02)**	0.27 (0.02)	**0.24** **(0.02)**	0.18 (0.03)	**0.23** **(0.02)**
Log-likelihood	1464.318	−1522.579	−1536.126	−1637.811	−1522.322
*P* cf. Model 1	0.49	0.88	0.34	0.58	0.85
Model 3					
MZ twin pairs	0.38 (0.02)	0.36 (0.02)	0.33 (0.02)	0.24 (0.02)	0.33 (0.02)
DZ twin pairs	0.38 (0.02)	0.36 (0.02)	0.33 (0.02)	0.24 (0.02)	0.33 (0.02)
Sister pairs	0.38 (0.02)	0.36 (0.02)	0.33 (0.02)	0.24 (0.02)	0.33 (0.02)
Log-likelihood	1550.395	−1576.761	−1590.299	−1652.067	−1585.007
*P* cf. Model 1	10^−38^	10^−24^	10^−24^	10^−6^	10^−27^
*P* cf. Model 2	10^−39^	10^−25^	10^−25^	10^−7^	10^−29^
Model 4					
MZ twin pairs	0.69 (0.02)	**0.62** **(0.02)**	0.59 (0.02)	**0.39** **(0.03)**	0.59 (0.02)
DZ twin pairs	0.34 (0.02)	**0.31** **(0.01)**	0.29 (0.01)	**0.20** **(0.01)**	0.29 (0.01)
Sister pairs	0.34 (0.02)	**0.31** **(0.01)**	0.29 (0.01)	**0.20** **(0.01)**	0.29 (0.01)
Log-likelihood	−1468.855	−1524.455	−1539.647	−1638.075	−1526.902
*P* cf. Model 1	0.01	0.15	0.02	0.7	0.01
*P* cf. Model 2	0.003	0.05	0.008	0.5	0.002

The estimates for the best fitting models are shown in bold.

**Table 3 cancers-14-01483-t003:** Variance components for mammogram risk scores (standard errors in parentheses) for ACE (Model 1) and AE + CMZ (Model 3) with *p*-values for designated comparisons of model fits based on log-likelihoods.

Variance Components	*Cumulus*	*Cumulus-percent*	*Altocumulus*	*Cirrocumulus*	*Cumulus-white*
Model 1					
Additive genetic	0.69 (0.02)	**0.62** **(0.02)**	0.59 (0.02)	**0.39** **(0.03)**	0.59 (0.02)
Common environment	0 (NA)	**0** **(NA)**	0 (NA)	**0** **(NA)**	0 (NA)
Log-likelihood	−1468.86	**−1524.46**	−1539.65	**−1638.08**	−1526.90
Model 2					
Additive genetic	**0.56** **(0.05)**	**0.53** **(0.05)**	**0.48** **(0.02)**	0.36 (0.03)	**0.46** **(0.05)**
MZ-specific environment	**0.15** **(0.05)**	**0.10** **(0.05)**	**0.14** **(0.02)**	0.04 (0.03)	**0.15** **(0.05)**
Log-likelihood	−1464.32	**−1522.58**	−1536.13	−1637.81	−1522.32
*P* cf. Model 1	0.003	**0.05**	0.008	0.5	0.002

The estimates for the best fitting models are shown in bold. NA = not applicable because the parameter estimate had hit the bound of 0.

## Data Availability

Data available on request due t ethical reasons.

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
