# Peer review of "Familial Aspects of Mammographic Density Measures Associated with Breast Cancer Risk"

_cancers, 2022, doi:10.3390/cancers14061483_

Round 1

Reviewer 1 Report

Manuscript ID: cancers-1602614

Title: Familial aspects of mammographic density measures associated with breast cancer risk

Introduction

Cumulus, Cumulus-percent, Altocumulus, Cirrocumulus, and Cumulus-white are mammogram risk scores (MRSs) used to predict the risk of breast cancer through mammographic density refers to the white and bright areas on the mammographic image. To estimate how much MRSs were correlated in relatives and the proportion of variance due to genetic factors. The author investigated the MRSs from the 593 monozygotic (MZ) and 326 dizygotic (DZ) female twins and their sisters. The results demonstrated that the correlations for MZ twin pairs are in the range about 0.4 to 0.7, which means half or more of variance of these MRSs is familial. The MZ pair correlations are greater than DZ pairs and sister pairs. In conclusion, most genetic factors that explain one-quarter to one-half as much of familial aggregation of breast cancer that explained by the polygenic risk score.

Significance & Novelty

The previous study indicated that masking increases the risk of having cancer missed at mammographic screening, it reduces, not increases, the incidence of screen-detected breast cancer. Therefore, the Author defines a mammogram risk score (MRS) by mammographic density as the residual of a mammographic measure after adjusting for age and BMI based on analysis of controls. The MRSs from 593 monozygotic (MZ) and 326 dizygotic (DZ) female twins and their sisters were measured to investigate the correlation in relatives and the proportion of variance due to genetic factors. The study will provide the information about familial aggregation of breast cancer that is explained by the current best polygenic risk score.

Major weakness

  • Risk factor questionnaires were collected in this study including the information about smoking history, alcohol consumption, onset and cessation of menstruation, and use of hormone replacement therapy. The author needs to describe how to evaluate the correlation of MRS from the MZ pair, DZ pairs, and sister pairs and genetic variance but not other risk factors including the information about smoking history, alcohol consumption, onset, and cessation of menstruation, and use of hormone replacement therapy. Other risk factors are whether and how to be excluded in this study.
  • Author defines a mammogram risk score (MRS) by mammographic density as the residual of a mammographic measure after adjusting for age and BMI based on analysis of controls. Why do only two factors age and BMI being adjusted in mammogram risk score (MRS)? Author whether considering adjusting the other risk factors in the mammogram risk score (MRS).
  • Author defines a mammogram risk score (MRS) by mammographic density after adjusting for age and BMI based on analysis of controls. The author must clearly describe how to define the analysis of controls in this study.
  • The other non-genetic factors such epigenetic factors and environmental factors associated with risk of breast cancer need to discuss in this study.
  • The author mentions that comparison of the proportion of familial aggregation in breast cancer explained by these MRS with that of the latest polygenic risk factor (PRS). Previous study suggested that multiple polygenic risk scores (PRSs) for breast cancer have been developed from large research consortia; however, their generalizability to diverse clinical settings is unknown. How do authors define meaningful polygenic factor variance that may contribute to the familial aggregation of breast cancer risk.

Minor weakness

  • In the text of material and methods, the sub-title must be consistent.
  • Author mention sample data collected from Australian Mammographic Density Twins and Sister Study. Does the author consider that people of different races and regions have different polygenetic variances in the familial aggregation of breast cancer?

Author Response

Major weakness

  • Risk factor questionnaires were collected in this study including the information about smoking history, alcohol consumption, onset and cessation of menstruation, and use of hormone replacement therapy. The author needs to describe how to evaluate the correlation of MRS from the MZ pair, DZ pairs, and sister pairs and genetic variance but not other risk factors including the information about smoking history, alcohol consumption, onset, and cessation of menstruation, and use of hormone replacement therapy. Other risk factors are whether and how to be excluded in this study.
  • Author defines a mammogram risk score (MRS) by mammographic density as the residual of a mammographic measure after adjusting for age and BMI based on analysis of controls. Why do only two factors age and BMI being adjusted in mammogram risk score (MRS)? Author whether considering adjusting the other risk factors in the mammogram risk score (MRS).

Thank you for pointing this issue out. We had previously found that risk factors other than age and BMI explained very little variance (at most a few percent) in the conventional density measure, Cumulus (Nguyen et al.,2013). This was trivial compared with the variance explained by age and BMI. We found that the same applies to the new density measures, Altocumulus and Cirrrocumulus, but had not mentioned this. We have therefore added at the end of the first paragraph in the Statistical Methods:

“(We had previously found that breast cancer risk factors measured by questionnaire other than age and BMI explained at most a few percent of variance in the conventional density measure, Cumulus, and this was trivial compared with the variance explained by age and BMI [2]. We have also now found that the same applies to the new density measures, Altocumulus and Cirrrocumulus.)”

  • Author defines a mammogram risk score (MRS) by mammographic density after adjusting for age and BMI based on analysis of controls. The author must clearly describe how to define the analysis of controls in this study.

We have elaborated on the text to further explain how the residuals were calculated by adding to the first paragraph:

“This was performed using linear regression applied to data for controls only to estimate the two regression coefficients and the constant term and thereby the expected value as a function of age and BMI and the variance of the residuals for controls. For both cases and controls, their observed transformed values were then subtracted from their expected values based on the regression analysis of controls above to derive the residuals, which were then divided by the variance of the residuals for controls.

  • The other non-genetic factors such epigenetic factors and environmental factors associated with risk of breast cancer need to discuss in this study.

There is little replicable evidence that epigenetic factors are risk factors for breast cancer and we only have epigenetic data for about 20% of the sample. We have already published that we found no association between epigenetic factors and conventional density, Cumulus; see Li S et al. Genome-wide association study of peripheral blood DNA methylation and conventional mammographic density measures. Int J Cancer 2019;145:1768-1773. We are continuing to work on these issues.

  • The author mentions that comparison of the proportion of familial aggregation in breast cancer explained by these MRS with that of the latest polygenic risk factor (PRS). Previous study suggested that multiple polygenic risk scores (PRSs) for breast cancer have been developed from large research consortia; however, their generalizability to diverse clinical settings is unknown. How do authors define meaningful polygenic factor variance that may contribute to the familial aggregation of breast cancer risk (sic).

The generality of the current PRS to diverse populations is beginning to become clear, and a recent paper found that the PRS for white Caucasian women applies quite well to South-east Asian women, with the risk gradient being attenuated by only 10-20% (REF). We have based our comparison of variances by using the strongest PRS known to date so as not to over-inflate our statements about the familial strength of the mammogram risk scores.

 Minor weakness

 In the text of material and methods, the sub-title must be consistent.

 We have corrected the sub-titles

  • Author mention sample data collected from Australian Mammographic Density Twins and Sister Study. Does the author consider that people of different races and regions have different polygenetic variances in the familial aggregation of breast cancer?

Participants in the AMDTSS are about 90% Caucasian, being born in Australia, New Zealand, the British Isles or Western and Southern Europe. We have commented on the differences in PRS by ethnicity above.

Reviewer 2 Report

Comment 1.

Among the 593 monozygotic (MZ) and 326 dizygotic (DZ) female twin pairs and 1,592 of their sisters in the authors' research, dataset for their genetic variants in breast cancer-associated genes should be compared with the mammogram risk scores (MRSs). 

Author Response

Response to Reviewers Comments (in italics)

Comment 1.

Among the 593 monozygotic (MZ) and 326 dizygotic (DZ) female twin pairs and 1,592 of their sisters in the authors' research, dataset for their genetic variants in breast cancer-associated genes should be compared with the mammogram risk scores (MRSs).

We agree, and this is what our next paper is about. It is called “Genetic aspects of mammographic density measures associated with breast cancer risk” and is soon to be submitted to the same journal; see Reference 14. Hopefully the two papers can appear together.

Reviewer 3 Report

the authors of the present study titled "Familial aspects of mammographic density measures associated with breast cancer risk" use familial correlations in the MRS that could have a genetic cause to explain familial aggregation of breast cancer. Following are some questions regarding the background of the study:

  1. Introduction, lines 89-90: the term 'respectively' is used here. Do the authors mean the three characteristics are different for Altocumulus and Cirrocumulus? Or do all three characteristics (the transformed, adjusted and standardized MRSs) define both MRSs?
  2. Introduction, line 104: By "s they were about one-half of this for dizygotic (DZ) twin pairs" do the author mean that correlations for DZ were 0.5 or 0.3?

Author Response

Response to Reviewers Comments (in italics)

  1. Introduction, lines 89-90: the term 'respectively' is used here. Do the authors mean the three characteristics are different for Altocumulus and Cirrocumulus? Or do all three characteristics (the transformed, adjusted and standardized MRSs) define both MRSs?

We moved “respectively” to the end of the sentence. There is one MRS for each measure. We added: “Therefore, Cumulus is the MRS for Cumulus, Altocumulus is the MRS for Altocumulus. and Cirrocumulus is the MRS for Cirrocumulus.”

  1. Introduction, line 104: By "they were about one-half of this for dizygotic (DZ) twin pairs" do the author mean that correlations for DZ were 0.5 or 0.3?

 We mean 0.3 – we have inserted “(about 0.3)” to clarify this.

Round 2

Reviewer 1 Report

Manuscript ID: cancers-1602614

Title: Familial aspects of mammographic density measures associated with breast cancer risk

The authors have answered most of the questions I addressed in the satisfactory manner. It should be OK for publishing in Cancers except one minor change may be made prior to publication.

  • The author must check the format of the Materials and Methods

2.1 Sample

2.2 Risk Factor Questionnaire.

Reviewer 2 Report

Comment 1.

If the authors honestly guarantee that their next paper includes such genetic variants underlying their discovery in the present paper, I support the publication of this manuscript in this Journal.